



**Temporary confined water responsible for triggering the landslide of a piedmont gentle slope in Ningzhen**
**Area, China**
Shulan Guo[1], Yang Liu[1,2], Changhong Yan[1], Liangchen Yu[1], Baotian Liu[3], Zezheng Gao[1]
[1]School of Earth Sciences and Engineering, Nanjing University, Nanjing, Jiangsu, China
[2]Guangzhou Urban Planning & Design Survey Research Institute, Guangzhou, Guangdong, China
[3]No. 3 Geological Party of Jiangsu Bureau of Geology and Mineral Resource, Zhenjiang, Jiangsu, China
**Corresponding author**
Changhong Yan
Nanjing University
No.163 Xianlin Avenue, Nanjing
210023 Jiangsu Province
PR China
Email: yanchh@nju.edu.cn
Phone: +86 13952091802

**Abstract**
The traditional viewpoint is that a piedmont gentle slope is conducive to the overall stability of a mountain, which in
turn has a counterpressure effect on the whole mountain. However, in recent years, with the increase in extreme
heavy rainfall, some landslides have occurred in the piedmont gentle slopes in Ningzhen area. On July 6, 2016, a
landslide occurred on the P0 slope of Paomashan Mountain, which is a typical piedmont gentle slope. After field
investigation and analysis of monitoring data, we found that it was an intermittent creeping landslide, staged by
initial deformation, isokinetic deformation and accelerated deformation. Survey data show that the slope has a very
special stratum structure, that is, clay in the surface, coarse-grained soil or weathered rock in the middle, and
bedrock in the bottom. In addition, the permeability of each layer is weak-strong-weak from top to bottom.
According to the monitoring data, we found that temporary confined water (confined water formed and dissipated in



a certain period of time that has never been encountered before) in the middle layer of the slope was the most
important factor in inducing the landslide. Through numerical simulation, we analyzed the formation process and
influencing factors of the temporary confined water. Finally, we propose effective control measures for this kind of
landslide. The research results can be used in the treatment of similar piedmont gentle slope landslides in the
Ningzhen area of China.

**Keywords**: piedmont gentle slope; temporary confined water; intermittent creeping landslide; numerical simulation;
control measures

**1. Introduction**

According to the traditional view, a piedmont gentle slope is beneficial to the overall stability of a mountain

(Zhou and Ou 1997; Pánek et al. 2014; Yan et al. 2019), which in turn has a counterpressure effect on the whole
mountain. However, in recent years, landslides have increased significantly in piedmont gentle slopes in the
Ningzhen area (Nanjing-Zhenjiang area in Jiangsu Province, China). For example, on July 3, 2016, a large-scale
landslide occurred at Youzi Mountain, which is a typical piedmont gentle slope in Nanjing; on October 25, 2016, a
landslide occurred on a piedmont gentle slope in the Fangshan scenic area of the Jiangning district, Nanjing, which
led to the closure of the scenic area for several months. Some scholars believe that the occurrence of these landslides
has a strong relationship with rainfall (Lo et al. 2010; Bai et al. 2013; Yu et al. 2020). Simultaneously, some studies
have been published on the characteristics and sliding mechanisms of piedmont gentle slope landslides. Trandafir et
al. (2013) employed numerical analysis to illustrate two major concepts that address the geomechanics of
catastrophic landslides on gentle slopes in liquefiable soils due to earthquakes. Taking one landslide of Japan as an
example, they believed that the gentle slope may experience large landslide movements due to earthquake-induced
ground liquefaction as a result of void redistribution and formation of water films in liquefied deposits with
continuous low-permeability interlayers. Adopting an early warning system called MoniFLaIR, Capparelli and
Tiranti (2010) monitored and analyzed the influence of rainfall on landslides in the Piedmont area (Northern Italy)
(Capparelli and Tiranti 2010; Trandafir et al. 2013).

Confined water is formed between two impermeable aquifers and has confined properties. If the surface of

the slope is an impermeable layer and the middle is a permeable layer, high-intensity rainfall may not allow the


converged groundwater to be discharged immediately along the interface, thus forming a "confined basin"(Huang et
al. 2005; Jiao et al. 2005; Vennari et al. 2014). As shown in Fig. 1, the mountains in the Ningzhen area show a
special characteristic: The upper part of the mountain is mainly exposed bedrock with fracture development, with a
steep slope, generally between 30° and 50°; the lower part of the mountain is composed of Quaternary gravelly soil
and loose sediments with slope angles mostly between 8° and 15°, which we refer to as a piedmont gentle slope. The
piedmont gentle slope in the Ningzhen area has the topographic structure of a steep upward and gentle downward
with a stratum structure: clay in the surface, coarse-grained soil or weathered rock in the middle and bedrock in the
bottom, which forms a special "binary stratigraphic structure" (Yan et al. 2019). The permeability of the strata is
weak-strong-weak from top to bottom. **Additionally,** the upper part of the slope is usually rock with weathered
fissures, which is convenient for rainfall infiltration. This slope structure is conducive to the formation of confined
water (Mikoš et al. 2004; Yan et al. 2010; Zeng 2010).

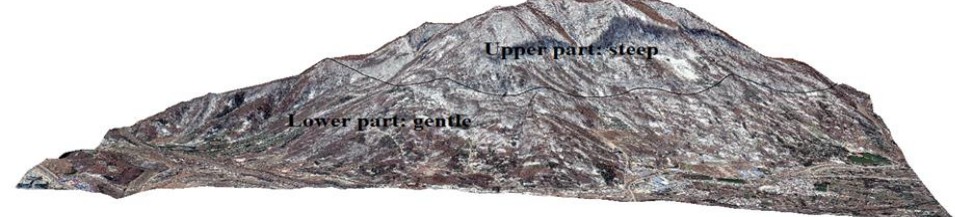


**Fig. 1** Schematic diagram of the mountain shape in Ningzhen Area
In 2015, 7 landslides occurred in the upper part of the mountain on the western and northern sides of
Paomashan Mountain (Fig. 2). Nearly 4 million yuan was spent for treatment, and the reinforcement measures of
antislide piles and bolt lattice were set up to control the sliding effectively. But the lower part of the slope was not
treated because they believed that it is relatively gentle and no landslide would occur. We set up two monitoring
holes in the lower part of the mountain to monitor the displacement and water level of the mountain just in case. In
June 2016, the lower slope of P0 experienced downhill scarps and tension cracks at the rear edge of the slope, as
shown in Fig. 3, causing the antislide pile in the upper part of the mountain to experience cracks and causing the soil
of the slope to move significantly downward. According to the monitoring data, we found that temporary confined
water (confined water formed and dissipated in a certain period of time) is the most important factor in triggering
landslides. However, there are very few studies on temporary confined water and its influence on the stability of
piedmont gentle slopes.



(a)                                    (b)

**Fig. 2** Geographical location of the research area    (The map in (a) is from ⓒ Google Earth)

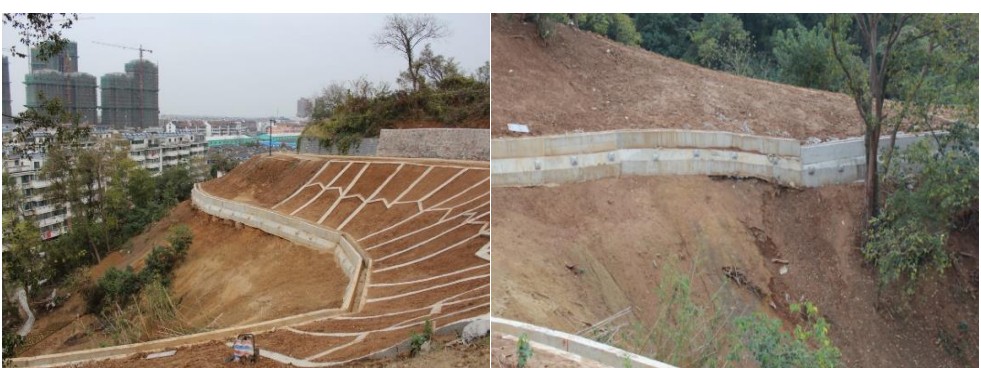


(a)  Original supporting structure                    (b) Cracks in antislide piles

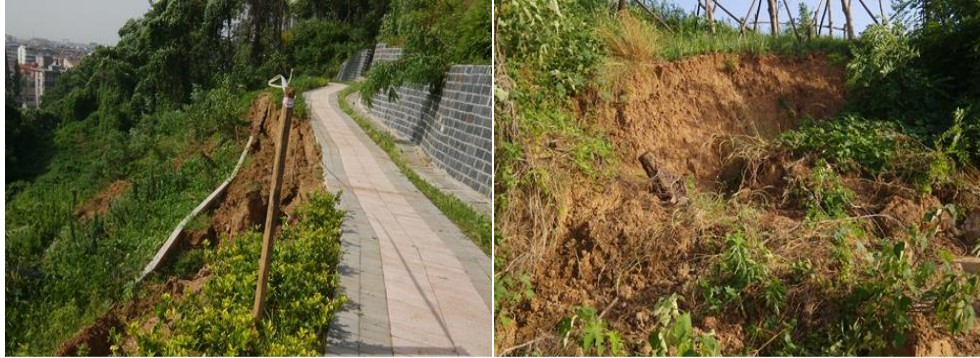


(c) Downhill scarps and tension cracks at the rear edge of the slope

**Fig. 3** Profile of the P0 slope

**2. Background**



Zhenjiang city is located in the south of Jiangsu Province, adjacent to Nanjing. There are many low

mountains and residual hills in the city. It belongs to the subtropical monsoon climate zone, with a high temperature
and rainy climate in the summer. The rainy season is concentrated in June, July and August. Paomashan Mountain is
located in the center of Runzhou District, Zhenjiang City, Jiangsu Province (Fig. 2), which is a typical soil slope.
According to the field survey data, the formation lithology in this slope is roughly divided into three layers, from top
to bottom (Fig. 4): silty clay in the surface, gravel in the middle with relatively developed fissures, strongly
weathered porphyry quartz diorite in the bottom, mainly composed of plagioclase, quartz, hornblende, etc. The
physical and mechanical properties are shown in Table 1.

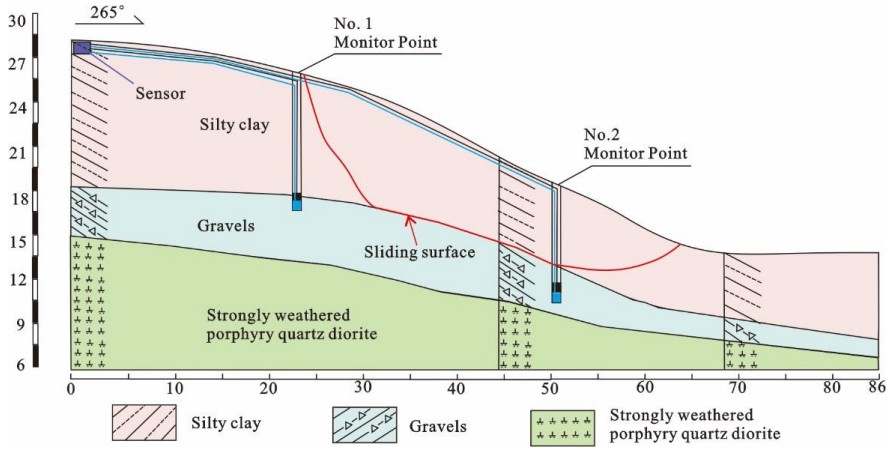


**Fig. 4** The geological map of 1-1' section
*2.1. Overview of the landslide*

Since June 20, 2016, there has been continuous heavy rainfall in Zhenjiang city. At approximately 10 a.m.

on June 24, confined water was detected. At 5:00 p.m. on June 25, the monitoring system began to receive
displacement data. On July 6, the landslide happened, which was a bedding slip (Fig. 4). The displacement data of
the 12-day monitoring period are shown in Fig. 5. The change in water pressure is shown in Fig. 6.

According to the field survey, the slide body is located in the middle and lower part of the slope body. With

creep deformation of the slope body, subsequent deeper and wider cracks were gradually formed at 0.2 m below No.
1. Under the condition of continuous heavy rainfall, the tension crack continued to extend, cut through and form a
down bench, which provided an effective free surface for the soil above. The maximum displacement of No. 2 is
1337 mm, while the maximum displacement of No. 1 is 281 mm (Fig. 5). The sliding speed of No. 1 is obviously




smaller than that of No. 2 (Table 2). The displacement-time relationship curve in the process of the landslide
conforms to the characteristics of typical intermittent creeping landslides (Xu et al. 2008; Tang et al. 2014).

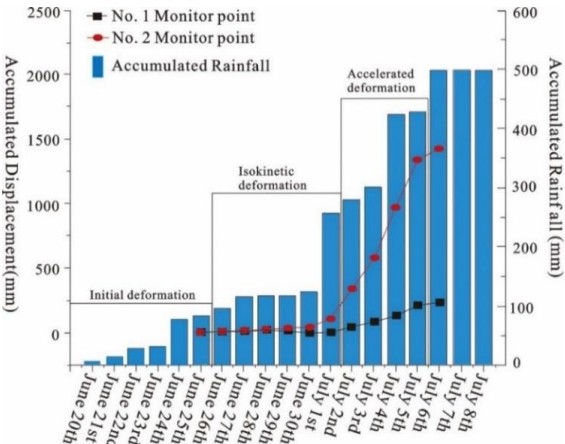


**Fig. 5** Relative relationship between rainfall and displacement

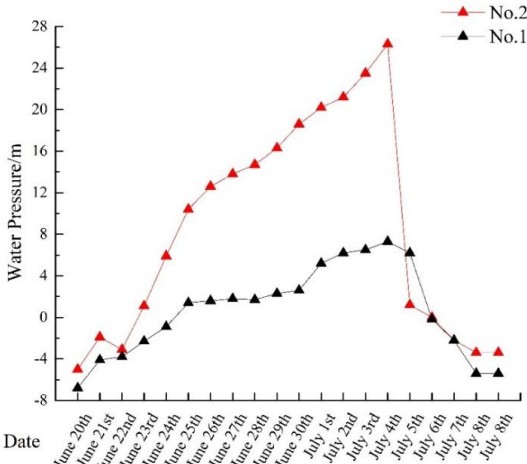


**Fig. 6** Change in the water pressure





**Table 1.** The physical and mechanical properties of the soil layers

| Layer Soil Type | Unit Weight (kN / m³) | | Cohesive force (kPa) | Internal friction angle (°) | Compression modulus (MPa) | Poisson's ratio (μ) | Plasticity index | Permeability coefficient |
|---|---|---|---|---|---|---|---|---|
| | Natural | Saturated | | | | | | |
| Silty clay | 18.0 | 19.7 | 36.7 | 6.1 | 6.1 | 0.3 | 14.6 | $1.5 \times 10^{-5}$ |
| Gravels | 18.5 | 20.1 | 65.3 | 10.2 | 10.6 | 0.28 | 16.1 | $2.5 \times 10^{-2}$ |
| Strongly weathered porphyry quartz diorite | 27.3 | 27.3 | 38.0 | 54.0 | 7000 | 0.20 | -- | $1 \times 10^{-6}$ |

**Table 2.** Average slip velocity of monitoring point (mm/h)

| Time | June | | | | | | July | | | | | |
|---|---|---|---|---|---|---|---|---|---|---|---|---|
| | 25 | 26 | 27 | 28 | 29 | 30 | 1 | 2 | 3 | 4 | 5 | 6 |
| No. 1 | 0.25 | 0.13 | 0.04 | 0.11 | 0.08 | 0.23 | 1.25 | 1.98 | 2.27 | 2.67 | 1.56 | 0.42 |
| No. 2 | 0.75 | 0.42 | 0.19 | 0.35 | 0.35 | 1.33 | 6.83 | 11.77 | 16.17 | 13.97 | 3.93 | 0.00 |



The landslide process can be divided into three stages: initial deformation, isokinetic deformation and
accelerated deformation.
*2.1.1. Initial deformation*
From June 20 to 25, continuous rainfall occurred in Zhenjiang, during which the rainfall reached 45.2 mm
on the 24th. According to the actual monitoring displacement data, the initial deformation of the slope occurred at
16:00 on the 25th and was 11 mm, the initial deformation speed was 2.1 mm/h, and the initial acceleration was 0.27
mm/h$^2$.
*2.1.2. Isokinetic deformation*
Continuous rainfall increased the water content in the slope, and the sliding force gradually increased. On
June 26, the slope entered the isokinetic deformation stage, and the deformation gradually increased. The average
deformation rate was 0.4 mm/h, and the acceleration range was -0.01~0.01 mm/h$^2$. The number of subsequent
deeper and wider cracks was obviously increased, the length was continuously extended, and there was a trend of
gradual penetration. There are shear dislocation zones in the gullies on the northern side of the slope body, and
several seepage points can be seen in the middle and lower part.
*2.1.3. Accelerated deformation*
From June 30 to July 1, there was continuous heavy rainfall, during which the rainfall on the 1st reached
88.8 mm, and the deformation of the slope increased significantly. At 8:00 on July 1, the slope entered the
accelerated deformation stage. In this stage, the deformation rate of the slope accelerated, and the deformation
increased to 11.3 mm/h. The range of acceleration fluctuated greatly (-0.40~0.52 mm/h$^2$). With the decrease in
rainfall, the acceleration gradually decreased to 0, and the landslide returned to the stage of isokinetic deformation.
However, at this time, the average deformation rate of the landslide was 13.9 mm/h, and the rainfall on the 4th
increased to 122.2 mm. The landslide again enters the stage of accelerated deformation. The acceleration increased
from 0 to 1.1 mm/h$^2$, and the deformation rate reached 22.3 mm/h. At this time, the sliding surface was completely
connected, and the slope was damaged by sliding.
**3. Influence of temporary confined water on landslide**
Fig. 6 shows that under continuous heavy rainfall, confined water is formed in the gravel layer in the
middle of the slope, with a maximum water pressure of 26.3 m. However, with the decrease in rainfall and the
pressure release effect of slope deformation, the water pressure will gradually decrease and finally dissipate, which




we refer to as the "temporary confined aquifer". According to the traditional view, circular sliding generally occurs
in the soil slope (Yan et al. 2016), but the study landslide is a bedding landslide because of the temporary confined
water (Fig. 4).

On the one hand, due to the pressure of the confined aquifer, at the roof of the temporary confined water

(the bottom of the silty clay layer), the "uplift pressure" perpendicular to the contact surface is generated, the force
between the upper and the middle soil layers is weakened, the sliding power is increased, and the antisliding power
is reduced so that the stability of the slope is reduced; on the other hand, the silty clay with gravel layer is enriched
when the confined water is collected, the weight of the slope increases greatly, and the sliding power generated in
the lower part of the slope also increases greatly. At the same time, under the combined action of the "uplift
pressure" of the confined water and the seepage force of the middle layer, the shear failure to the initial crack point
of the landslide will occur at the slope toe. This accelerates the sliding of the slope, and the uplift pressure of the
temporary confined water on the soil layer interface weakens the interaction between the soil layers, thus making the
slope slide along the layer.
*3.1. Numerical simulation analysis*

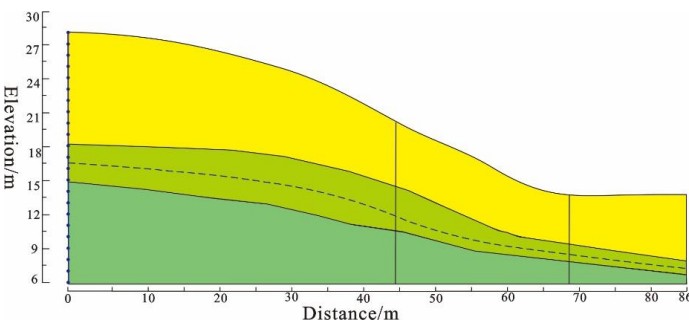


**Fig. 7** The model

Combined with the above analysis, it can be seen that temporary confined water is the main factor

triggering the landslide. To verify this analysis, using the Seep/W modular of Geo-studio software as shown in Fig.
7, we simulated the change in groundwater level under different rainfall conditions and observed the formation of
confined water. We set the surface as the rainfall infiltration boundary. The permeability of the middle layer and the
upper layer is considerably different, and the rainfall duration is relatively short; thus, the upper soil layer can be
regarded as the impermeable boundary. Moreover, the rainfall infiltration boundary is only set at the exposed part of
the middle layer, and the slope has a certain slope so there is no ponding, and the slope surface should be regarded as



a zero-pressure head. When the pressure head is greater than 0 m, confined water is formed. The physical and
mechanical parameters of the layers are shown in Table 1.

### 3.2. Formation of temporary confined water

#### 3.2.1. Effect of rainfall intensity and rainfall duration

To study the influence of intensity and duration of rainfall on the confined water level, a steady-state flow

is taken as the initial state of groundwater seepage. The rainfall intensity was set to 75 mm/d, 100 mm/d, 125 mm/d,
150 mm/d and 175 mm/d, and the change in the temporary confined water level was observed within 3 d of rainfall.
The relationship between the pressure head and the horizontal distance at the bottom of the confining bed is shown
in Fig. 8.

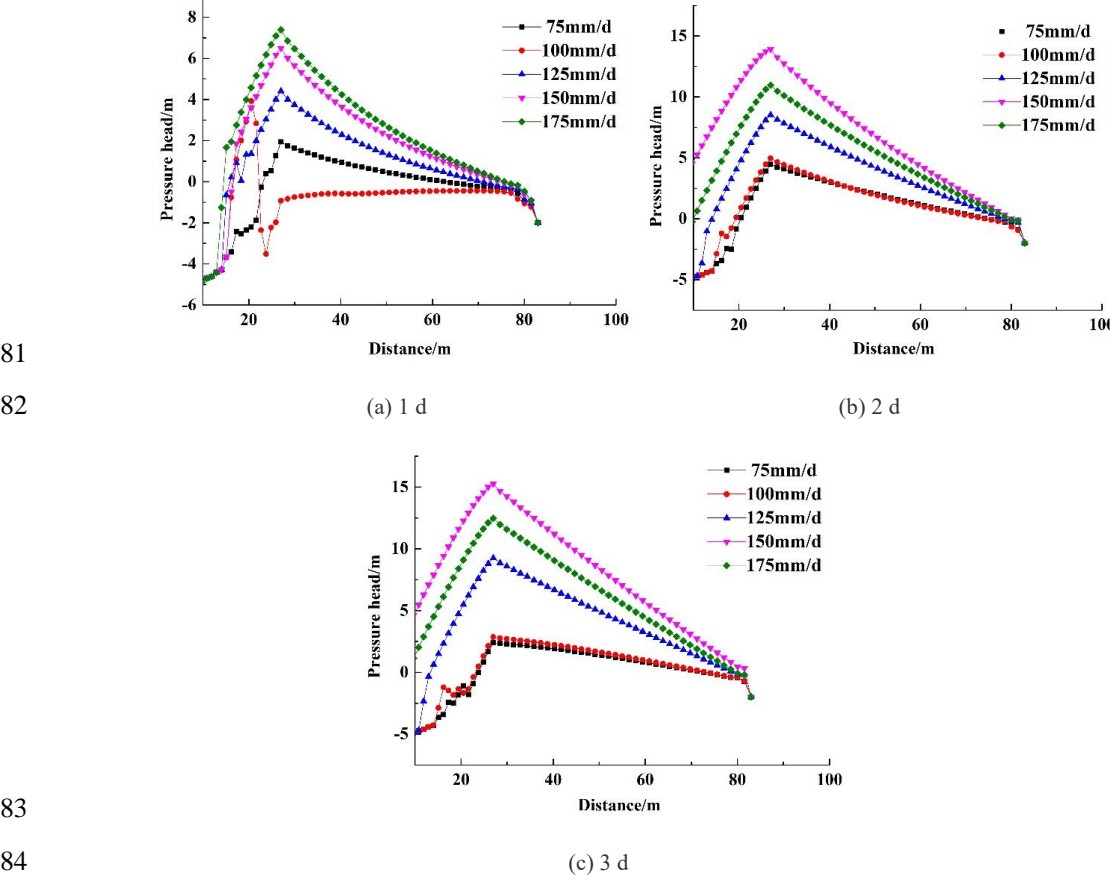


(a) 1 d                                    (b) 2 d

                                (c) 3 d

**Fig. 8** Relationship between confined water level and rainfall intensity

The change in the pressure head in the horizontal direction can be roughly divided into unconfined areas


and confined areas. In the initial stage of rainfall (< 1 d), the variation in confined water level increases with the
increase in rainfall intensity (except for the 75 mm/d rainfall). In the middle stage of rainfall (1 d~2 d), when the
rainfall intensity is 150 mm/d instead of 175 mm/d, the confined water level reached a maximum of 13.5 m. In the
later stage of rainfall (>2 d), the confined water level reached a maximum of 15 m. Therefore, 150 mm/d is the
optimal rainfall intensity for the formation of confined water.

In addition, rainfall duration is also an important factor affecting the confined water level. Based on the

rainfall data in the Ningzhen area, taking the rainfall intensity of 125 mm/d and 150 mm/d as examples, the effects
of rainfall duration on confined water level are studied (Fig. 9).

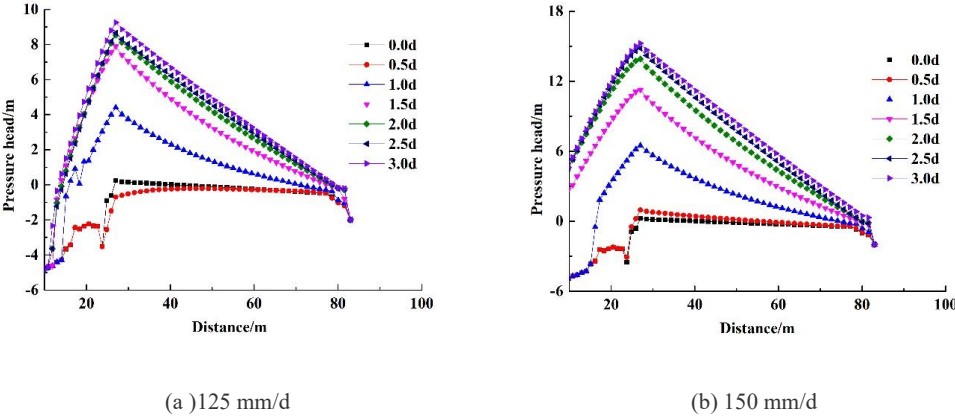


(a )125 mm/d                       (b) 150 mm/d

**Fig. 9** Relationship between confined water level and rainfall duration

It can be seen from Fig. 9 that (1) When the rainfall intensity is small (＜125 mm/d): In the early stage of

rainfall (＜1.0 d), which is influenced by the stagnant water in the unsaturated zone, the supply of the middle and
lower parts of the slope is reduced, and the groundwater level decreases slightly; in the middle stage of the rainfall
(1.0 d ~ 2.0 d), the stagnant water in the unsaturated zone dissipates to the middle and lower parts of the slope, and
the confined water level increases rapidly; in the later stage of the rainfall (＞2.0 d), the confined water level
increases rapidly and gradually tends to be stable. (2) When the rainfall intensity is larger (>125 mm/d): In the early
stage of rainfall (< 1.0 d), the influence of the stagnant water in the unsaturated zone is weaker. The level of
groundwater shows a rising trend and forms confined water rapidly; in the middle stage of rainfall (1.0 d ~ 2.0 d), it
is the fastest stage of the confined water level increasing, and the increasing rate of confined water level decreases
gradually; in the later stage of rainfall (> 2.0 d), the water level tends to stabilize gradually. When the middle layer is





completely filled with rainwater, the groundwater seepage will enter the stable seepage stage. According to the
lifting speed and height of confined water, when the rainfall duration is approximately 2.0 d, it is beneficial to form
a higher confined water level.
*3.2.2. Effect of the permeability coefficient of the middle layer*
The permeability of the middle layer has a crucial influence on the formation of confined water (Finlay et
al. 1997; Jiao et al. 2005; Rosone et al. 2018). Therefore, we studied the confined water level under the condition of
different permeability coefficients of the middle layer when the rainfall intensity is 150 mm / d and the rainfall
duration is 2 days.

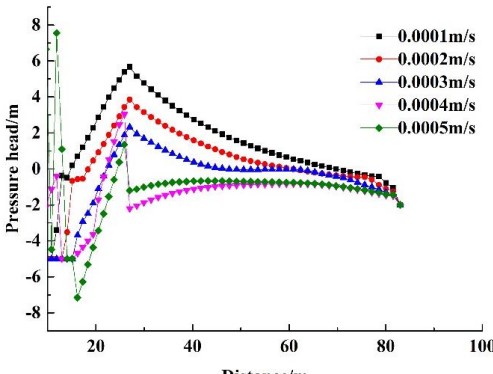


**Fig. 10** Relationship between confined water level and permeability coefficient
Fig. 10 shows that when the permeability coefficient is 0.0001 m/s, 0.0002 m/s and 0.0003 m/s, confined
water is formed in the middle and lower parts of the slope, and the confined water level decreases with the increase
in the permeability coefficient. When the permeability coefficient is 0.0004 m/s and 0.0005 m/s, only partially
confined water is formed in the upper part of the slope. Therefore, when the permeability coefficient of the middle
layer is between 0.0001 m/s and 0.0003 m/s, it is beneficial to the formation of confined water.
***3.3. Influence of temporary confined water on slope stability***
In addition, we use the limit equilibrium theory to analyze the stability of the P0 slope, and the results are
shown in Fig. 11. The stability coefficient of the slope is far greater than 1 when there is no strong rainfall, and the
slope is in a relatively stable state, and this indicates that strong rainfall is a necessary condition to induce such slope
sliding. According to Fig. 6, the stability coefficient of the slope is positively related to the water pressure level in
the slope. With the increase in rainfall duration and accumulated rainfall, the water pressure level of the slope



increases gradually, and the stability coefficient of the slope shows a significant downward trend.

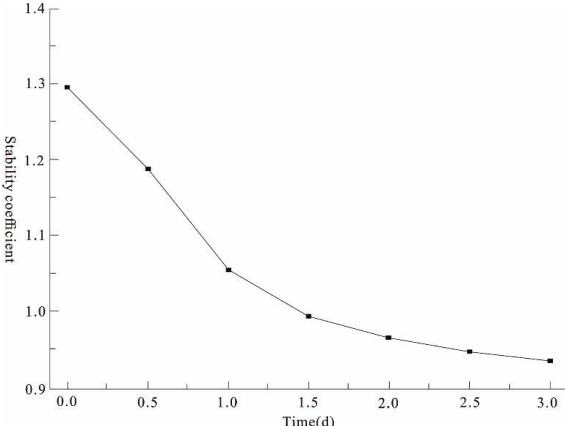


**Fig. 11** Relationship between slope stability coefficient and rainfall duration

**4. Discussion**

The main inducement of the piedmont gentle slope landslide is temporary confined water. The prevention

and control of this kind of slope should be based on water control: mainly interruption, drainage and diversion,
supplemented by blocking; in addition, corresponding engineering measures should be adopted, mainly antislide pile
and anchor cable. At the same time, different measures can be taken in different parts of the slope to prevent and
control the landslide.

For the upper part of the slope: as a rainfall infiltration channel, most of the rainwater penetrates into the

slope through this place. Intercepting peripheral water and drainage ditches can be set on the stable slope surface 5
m away from the exposed part of the gravel soil layer to prevent rainfall from converging on the slope surface; in
addition, the gravel soil exposed on the upper part of the slope can be replaced with clay and other materials with
poor permeability to weaken the permeability of this part and reduce rainwater infiltration.

For the middle part of the slope: temporary confined water in this part will produce an uplift effect

perpendicular to the bottom of the aquiclude. To weaken the effect of confined water, a dewatering well should be
installed in the middle of the slope. During continuous heavy rainfall, the groundwater level is first monitored by the
dewatering well. When the pressure head in the monitoring well reaches a predetermined dangerous value (the value
can be calculated by the slope stability), the slope will be dewatered by the dewatering well so that the groundwater
level in the monitoring well will always be kept within a certain safety range.



For the lower part of the slope: it is a collection area of groundwater. Drainage measures can be taken here
to reduce the highest water level in the slope. In addition, the self-weight of the soil mass and the thrust of the upper
part of the slope make the lower part of the slope bear more stress. Antislide piles can be built here to ensure the
safety of the slope.

**5. Conclusions**
Temporary confined water is a kind of confined water formed in the piedmont gentle slopes of the
Ningzhen area under extreme rainfall conditions. It has certain pressure properties and is affected by rainfall
duration, rainfall intensity and the permeability coefficient of the middle layer. Taking the landslide of P0 in
Paomashan Mountain in Zhenjiang City as an example, the sliding mechanism was studied. The following
conclusions can be drawn:
(1)    The piedmont gentle slope in the Ningzhen area has a special stratum structure, that is, clay in the

surface, coarse-grained soil or weathered rock in the middle, and bedrock in the bottom, and the

permeability of the layers is weak-strong-weak from top to bottom.

(2)    The landslide of P0 is an intermittent creeping landslide, which can be divided into three stages: initial

deformation, isokinetic deformation and accelerated deformation. Temporary confined water is a

necessary condition to trigger this type of landslide, and the slope stability coefficient will decrease

gradually with the increase in the confined water level.

(3)    According to the simulation results, when the rainfall intensity is 150 mm/d, the rainfall duration is

approximately 2.0 d, and the permeability coefficient of the middle layer is between 0.0001 m/s and

0.0003 m/s, which is beneficial to the formation of confined water.

(4)    For the prevention and control of landslides on gentle slopes in front of mountains in the Ningzhen

area, the principles of "water control" and "engineering prevention and control" should be followed.

Slope water control is usually based on interception, drainage and diversion, supplemented by

blocking; engineering measures are mainly based on antislide piles and anchor cables. At the same

time, different measures can be taken in different parts of the slope to prevent and control piedmont

gentle slope landslides.




**Acknowledgments**
This work was supported by the Natural Science Foundation of Jiangsu Province, China [grant number BE2019075].
Fieldwork was supported by the No. 3 Geological Party of Jiangsu Bureau of Geology and Mineral Resource. The
authors acknowledge Wei wang and Baotian Liu for on-site scheduling.
**Author contributions**
LY and YCH designed the study and performed the experiments;GSL and YLC analyzed the data, and wrote the
manuscript. XBT, LBT and GZZ carried out the field work.

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
