# Peer review of "Area, China 3 4 Shulan Guo1, Yang Liu1,2, Changhong Yan1, Liangchen Yu1, Baotian Liu3, Zezheng Gao1 5 6 1School of Earth Sciences and Engineering, Nanjing University, Nanjing, Jiangsu, China 7 2Guangzhou Urban"

_Natural Hazards and Earth System Sciences, 2019_

## Referee Comment (RC1) · Anonymous Referee #1 · 7 Feb 2020

Manuscript title: Temporary confined water responsible for triggering the landslide of a piedmont gentle slope in Ningzhen Area, China

Dear Authors

This paper insists to have found that temporary confined water in the middle layer of the slope was the most critical factor triggering landslides at piedmont gentle slopes in Ningzhen area. Field investigations, monitoring analysis, and numerical analysis were conducted to support the insistence and to analyze other influencing factors. Speaking in a general review, this paper should be majorly revised because the quality and descriptions of the results of field monitoring results is low and does not sufficiently back up its aforementioned main insistence (that the confined water is a major factor of the landslide triggering). On the other hand, the deductions from the numerical analysis and stability analysis are not warranted to be meaningful for general applications. Therefore, **this paper is not acceptable to be published and I judged that the paper should be rejected.** In addition, there are many points that require the authors' clarification as follows:

1. The quality of technical English writing is too low to be published. Strong recommendation of getting a language editing service.

2. [Line 55-58] Is "impermeable aquifers" a correct expression? Aquifer is defined as an underground layer of water-bearing "permeable" rock, rock fractures or unconsolidated materials. Moreover, if the surface layer is impermeable, how can rainfall infiltrate into the middle layer by crossing the surface layer?

3. [Line 58 and Fig. 1] Is the Ningzhen area a broader territory including Zhenjiang city in which the Paomashan Mountain is located? Moreover, how is the mountain shown in Fig.1 related to your study area, the P0 landslide? If they are different places, there seems no significant meaning in Fig. 1. It is difficult to understand the areal relations between the place names that are mentioned in your manuscript. Detailed explanation on your study area with a regional-scale map that clarifies the aforementioned questions is needed.

4. [Fig .4] If the formation of the three layers is like Fig. 4 and the middle layer has a significantly higher permeability than the upper and lower layers, isn't is more reasonable to speculate that the groundwater, which is temporarily formed in the middle("gravel") layer, will be drained out along the bottom of the middle layer that is connected to the lower end of slope and a flat area (see Fig. 4) ? Please justify how the water is confined and thereby pore-water pressure can be increased in the gravel layer of such a stratum structure.

5. [Line 73-74] Detailed information and descriptions on the monitoring system should be needed; what specific devices (or sensors) were used to monitor the displacement and water level at which specific depths (please indicate and mark at the specific depths of sensing in each of the two monitoring well in Fig. 4)? It is difficult to understand section 2.1 without the aforementioned information. For example, the monitoring system began to receive displacement data at what specific depths of which layer (Line 102-103)? The change of water pressure at what specific depths of which layer (Line 104)? The maximum displacements of No.2 and No.1 at what specific depths of which layer (Line 108-109)?

6. [Line 105-108] difficult to understand the paragraph. "Slide body", "slope body", and "sliding surface (in Fig. 4)"…………What does specifically each of the three terminologies indicate? What's the differences? Is the "subsequent and wider cracks" different from the sliding surface in Fig. 4? If so, why they are not depicted in Fig. 4?

7. [Line 111] The landslide conforms to the characteristics of "typical intermittent creeping landslides"……in what aspect? Is it typical for the type of creeping landslides to have such three distinct stages? Is the expression "intermittent" correct in the case of this landslide event? Most typical creeping landslides exhibit multiple reactivations in a long-term period. Please justify your expression with referencing to literatures.

8. [Table 1] Permeability coefficient ……Unit?

9. [Fig. 5] Please reconsider the period of Initial Deformation. It is mentioned that the initial deformation occurred at 16:00 on the 25[th] and was 11 mm. Therefore, the period should be from the starting time of the deformation to the time when the deformation first reached 11 mm. The boundary of Initial Deformation that is drawn seems improper.

10. [Line 131-134] It is written as if such phenomena as "increasing number of subsequent cracks" and "extending length and gradual penetration" were "visually observed" during the period of Isokinetic deformation. Clear evidences must be accompanied when the author insists that such phenomena took place during Isokinetic deformation.

11. [Line 133-134] shear dislocation zones, seepage points..........what's their relation with the Isokinetic deformation behavior? Evidences of these phenomena happening during the Isokinetic deformation?

12. [line 136-137] 88.8 mm............Is it correct amount? Seeing Fig. 5, rainfall seems to have accumulated as much as ~130 mm on the 1[st] July.

13. [Line 138-139] How come the deformation increased to 11.3 mm/h? The unit!

14. [Line 146-147 and Fig. 6] Why was the confined water not formed at the monitoring point 2 in the grave layer?

15. [Line 147] a maximum water pressure of 26.3 m ?? Unit!

16. [Line 150-151] Please theoretically justify landslide occurred bedding on the interface between the surface layer and the middle layer. If it was because of the increase in the pore-water pressure caused by confined water, isn't is more reasonable to see the sliding surface on the interface between the middle layer and the lowest layer?

17. [Line 155-156] "the silty clay ~ the weight of the slope increases greatly".........Again, if the authors want to insist that these unique phenomena happened, clear evidences or reasonable theoretical deduction must be suggested in advance.

18. [Line 157] the lower part of the slope......confusing and unclear expression. Where does it specifically indicate?

19. [Line 168-171] Where is the exposed part of the middle layer in Fig. 4 and Fig. 7? Why no indications in those figures? Moreover, rainfall infiltration boundary is supposed to be depicted as a series of arrows. Why no such indications in those figures? Such unkind descriptions exist a lot in this manuscript.

20. [Section 3.1] There is no explanation on how the initial condition (moisture or pore-water pressure state profiles) of the slope was set using what kind of analysis, what reference field data, or what kind of different boundary conditions....

21. [Line 179 and Fig. 8] 1) "the bottom of the confining bed".......Please clarify where this expression indicate by marking the nodes in Fig. 7 where the pore-water pressure data was picked up to plot Fig. 8. 2) Why do the plots in Fig. 8 exhibit sharp conversions of the gradient from positive to negative at some distances around 30 m?

22. [Line 186-187] "......can be divided into unconfined areas and confined areas."  ........Where do the authors mean by unconfined part and unconfined part? Another unkind description....

23. [Line 190-191] What is the reason of 150 mm/d reaching the highest water level instead of 175 mm/d? Furthermore, are these results of numerical study meaningful from the perspective of the applications in practice? In what aspect? Please persuade it.

24. [Line 198-210] It is very difficult to completely understand and match the descriptions in this paragraph with Fig. 9.

25. [Line 209] What does "lifting speed" mean?

26. [Line 218-209] the most optimal permeability 0.0001-0.0003................It is only a result deduced from

a single specific event condition. What aspect does this study have valuable meaning for a general application?

27. [Line 224] What kind of limit equilibrium theory and what equation? At which depth was the sliding surface set in the stability analysis? How did you apply the pore-water pressure term that changes over time into the limit equilibrium equation? Detailed descriptions on your assumptions and settings in the stability analysis is entirely omitted.....

28. [Line 227] Normally, we use "factor of safety" rather than "stability coefficient"....In addition, the stability coefficient is positively(?) related to the water pressure level? Not negatively?

---

## Author Comment (AC1) · 20 Feb 2020

The comment was uploaded in the form of a supplement:
https://www.nat-hazards-earth-syst-sci-discuss.net/nhess-2019-404/nhess-2019-404-AC1-supplement.pdf a certain period of time that has never been encountered before) in the middle layer of the slope was the most important factor in inducing the landslide. Through numerical simulation, we analyzed the formation process and influencing factors of the temporary confined water. Finally, we propose effective control measures for this kind of landslide. The research results can be used in the treatment of similar piedmont gentle slope landslides.

**Keywords**: piedmont gentle slope; temporary confined water; intermittent creeping landslide; numerical simulation; control measures

**1. Introduction**

According to the traditional view, a piedmont gentle slope is beneficial to the overall stability of a mountain (Zhou and Ou 1997; Pánek et al. 2014; Yan et al. 2019), which in turn has a counterpressure effect on the whole mountain. However, in recent years, landslides have increased significantly in piedmont gentle slopes in Jiangsu

Province, China. For example, on July 3, 2016, a large-scale landslide occurred at Youzi Mountain, which is a typical piedmont gentle slope in Nanjing; on October 25, 2016, a landslide occurred on a piedmont gentle slope in the Fangshan scenic area of the Jiangning district, Nanjing, which led to the closure of the scenic area for several months. Some scholars believe that the occurrence of these landslides has a strong relationship with rainfall (Lo et al. 2010; Bai et al. 2013; Yu et al. 2020). Simultaneously, some studies have been published on the characteristics and sliding mechanisms of piedmont gentle slope landslides. Trandafir et al. (2013) employed numerical analysis to illustrate two major concepts that address the geomechanics of catastrophic landslides on gentle slopes in liquefiable soils due to earthquakes. Taking one landslide of Japan as an example, they believed that the gentle slope may experience large landslide movements due to earthquake-induced ground liquefaction as a result of void redistribution and formation of water films in liquefied deposits with continuous low-permeability interlayers.

Adopting an early warning system called MoniFLaIR, Capparelli and Tiranti (2010) monitored and analyzed the influence of rainfall on landslides in the Piedmont area (Northern Italy) (Capparelli and Tiranti 2010; Trandafir et al. 2013).

Confined water is formed between two aquifuges and has confined properties. When the rain infiltrates into the lower part of the slope through the cracks in the upper part of the slope, if the surface of lower part of the slope is an impermeable layer and the middle is a permeable layer, high-intensity rainfall may not allow the converged

**Fig. 1.**

groundwater to be discharged immediately along the interface, thus forming a "confined basin"(Huang et al. 2005;
Jiao et al. 2005; Vennari et al. 2014). As shown in Fig. 1, the mountains in the Zhenjiang area show a special
characteristic: The upper part of the mountain is mainly exposed bedrock with fracture development, with a steep
slope, generally between 30° and 50°; the lower part of the mountain is composed of Quaternary gravelly soil and
loose sediments with slope angles mostly between 8° and 15°, which we refer to as a piedmont gentle slope. The
piedmont gentle slope in the Zhenjiang area has the topographic structure of a steep upward and gentle downward
with a stratum structure: clay in the surface, coarse-grained soil or weathered rock in the middle and bedrock in the
bottom, which forms a special "binary stratigraphic structure" (Yan et al. 2019). The permeability of the strata is
weak-strong-weak from top to bottom. Additionally, the upper part of the slope is usually rock with weathered
fissures, which is convenient for rainfall infiltration. This slope structure is conducive to the formation of confined
water (Mikoš et al. 2004; Yan et al. 2010; Zeng 2010).

[Figure]

**Fig. 1** Schematic diagram of Piedmont gentle slope in Zhenjiang area
In 2015, 7 landslides occurred on the western and northern sides of Paomashan Mountain (Fig. 2). Nearly 4
million yuan was spent for treatment, and the reinforcement measures of antislide piles and bolt lattice were set up
in the upper part of the slope to control the sliding effectively. But the lower part of the slope was not treated
because they believed that it is relatively gentle and no subsequent sliding would occur. Using inclinometer to
monitor the displacement of silty clay layer and digital water level gauge to monitor the change of water pressure of
gravel layer, they set up two monitoring holes in the lower part of the mountain just in case. In June 2016, the lower
slope of P0 experienced downhill scarps and tension cracks at the rear edge of the slope, as shown in Fig. 3, causing
the antislide pile in the upper part of the mountain to experience cracks and causing the soil of the slope to move

**Fig. 2.**

[Figure]

(c) Downhill scarps and tension cracks at the rear edge of the slope
**Fig. 3** Profile of the P0 slope
**2. Background**
Zhenjiang city is located in the south of Jiangsu Province, adjacent to Nanjing. There are many low
mountains and residual hills in the city. It belongs to the subtropical monsoon climate zone, with a high temperature
and rainy climate in the summer. The rainy season is concentrated in June, July and August. Paomashan Mountain is
located in the center of Runzhou District, Zhenjiang City, Jiangsu Province (Fig. 2), which is a typical soil slope.
According to the field survey data, the formation lithology in this slope is roughly divided three layers, from top
to bottom (Fig. 4): silty clay in the surface, gravel in the middle with relatively developed fissures, strongly
weathered porphyry quartz diorite in the bottom, mainly composed of plagioclase, quartz, hornblende, etc. The
physical and mechanical properties are shown in Table 1.

[Figure]

**Fig. 4** The geological map of 1-1' section
*2.1. Overview of the landslide*
Since June 20, 2016, there has been continuous heavy rainfall in Zhenjiang city. At approximately 10 a.m.
on June 24, confined water was detected. At 5:00 p.m. on June 25, the monitoring system began to receive
displacement data of the layer of silty clay and the change of water pressure of the layer of gravels. 
[revised manuscript text omitted]

was edited for proper English language, grammar, punctuation, spelling, and overall style by one or more of the highly qualified native English speaking editors at AJE.

This certificate was issued on **November 28, 2019** and may be verified on the AJE website using the verification code **671D-4227-5823-E669-6ED1**.

[Figure]

Neither the research content nor the authors' intentions were altered in any way during the editing process. Documents receiving this certification should be English-ready for publication; however, the author has the ability to accept or reject our suggestions and changes. To verify the final AJE edited version, please visit our verification page at aje.com/certificate. If you have any questions or concerns about this edited document, please contact AJE at support@aje.com.

AJE provides a range of editing, translation, and manuscript services for researchers and publishers around the world. For more information about our company, services, and partner discounts, please visit aje.com.

**Fig. 8.**

**Supplement:**

Dear Editors and Reviewers:

Thank you for your comments concerning our manuscript entitled "Temporary confined water responsible for triggering the landslide of a piedmont gentle slope in Ningzhen Area, China". Those comments are very helpful for revising and improving our paper. Based on the comments we received, careful modifications have been made to the manuscript. All changes were marked in red text. We hope the new manuscript will meet your magazine's standard. Below you will find our point-by-point responses to the comments/ questions:

Responds to the reviewer's comments:

Reviewer #1:

1. The quality of technical English writing is too low to be published. Strong recommendation of getting a language editing service

Response: Before submitting the manuscript, we have found a native English speaker to edit the manuscript. The editing certification will be attached for your reference. We also have a native English speaker double-checked the English for the revised R1 version.

2. [Line 55-58] Is "impermeable aquifers" a correct expression? Aquifer is defined as an underground layer of water-bearing "permeable" rock, rock fractures or unconsolidated materials. Moreover, if the surface layer is impermeable, how can rainfall infiltrate into the middle layer by crossing the surface layer?

Response: We are sorry for our misexpression in line 55-58, and we have modified the sentence. Additionally, we showed that the upper part of the slope is usually rock with weathered fissures, which is convenient for rainfall infiltration.

3. [Line 58 and Fig. 1] Is the Ningzhen area a broader territory including Zhenjiang city in which the Paomashan Mountain is located? Moreover, how is the mountain shown in Fig.1 related to your study area, the P0 landslide? If they are different places, there seems no significant meaning in Fig. 1. It is difficult to understand the areal relations between the place names that are mentioned in your manuscript. Detailed explanation on your study area with a regional-scale map that clarifies the aforementioned questions is needed.

Response: In line 58 and Fig. 1, we are very sorry for our incorrect writing, and we have modified the sentence and the Fig. 1.

4. If the formation of the three layers is like Fig. 4 and the middle layer has a significantly higher permeability than the upper and lower layers, isn't is more reasonable to speculate that the groundwater, which is temporarily formed in the middle("gravel") layer, will be drained out along the bottom of the middle layer that is connected to the lower end of slope and a flat area (see Fig. 4) ? Please justify how the water is confined and thereby pore-water pressure can be increased in the gravel layer of such a stratum structure.

Response: We are very sorry of our mistakes in expression which let you misunderstand. We have revised this figure. The thickness of the middle layer changes greatly, especially at the edge of the slope, the middle layer will gradually disappear. In addition, affected by the foundation engineering around the hillside, it is more difficult for water seepage.

5. [Line 73-74] Detailed information and descriptions on the monitoring system should be needed; what specific devices (or sensors) were used to monitor the displacement and water level at which specific depths (please indicate and mark at the specific depths of sensoring in each of the two monitoring well in Fig. 4)? It is difficult to understand section 2.1 without the aforementioned information. For example, the monitoring system began to receive displacement data at what specific depths of which layer (Line 102-103)? The change of water pressure at what specific depths of which layer (Line 104)? The maximum displacements of No.2 and No.1 at what specific depths of which layer (Line 108-109)?

Response: We agree with you very much and we have revised the sentence in line 74-75 、line 103-105 and Fig. 4. The fixed inclinometer is used for monitoring, which mainly monitors the displacement at the fixed depth showed in Fig. 4.

6. [Line 105-108] difficult to understand the paragraph. "Slide body", "slope body", and "sliding surface (in Fig. 4)"…………What does specifically each of the three terminologies indicate? What's the differences? Is the "subsequent and wider cracks" different from the sliding surface in Fig. 4? If so, why they are not depicted in Fig. 4?

Response: We are sorry to puzzled you and we have revised the misexpression in line 105-108 and Fig. 4. The "slide body" should be "slip body" meaning the part of rock and soil that has moved. The "slope body" should be "slope". subsequent and wider cracks

7. [Line 111] The landslide conforms to the characteristics of "typical intermittent creeping landslides"……in what aspect? Is it typical for the type of creeping landslides to have such three distinct stages? Is the expression "intermittent" correct in the case of this landslide event? Most typical creeping landslides exhibit multiple reactivations in a long-term period. Please justify your expression with referencing to literatures.

Response: Firstly, the landslide conforms to the characteristics of three distinct stages. Secondly, as mentioned above, there was a slide in 2015, but they believed that it is relatively gentle and no subsequent sliding would occur.

8. [Table 1] Permeability coefficient ……Unit?
Response: We have added it in the table 1.

9. [Fig. 5] Please reconsider the period of Initial Deformation. It is mentioned that the initial deformation occurred at 16:00 on the 25th and was 11 mm. Therefore, the period should be from the starting time of the deformation to the time when the deformation first reached 11 mm. The boundary of Initial Deformation that is drawn seems improper.

Response: Thank you for your suggestion and we have revised the boundary of Initial Deformation.

10. [Line 131-134] It is written as if such phenomena as "increasing number of subsequent cracks" and "extending length and gradual penetration" were "visually observed" during the period of Isokinetic deformation. Clear evidences must be accompanied when the author insists that such phenomena took place during Isokinetic deformation.

Response: We have revised the sentences in line 131-133.

11. [Line 133-134] shear dislocation zones, seepage points..........what's their relation with the Isokinetic deformation behavior? Evidences of these phenomena happening during the Isokinetic deformation?

Response: We have revised the sentences in line 131-133. And this is only to describe some phenomena of slope in the stage of Isokinetic deformation, not the evidences of these phenomena happening during the Isokinetic deformation.

12. [line 136-137] 88.8 mm............Is it correct amount? Seeing Fig. 5, rainfall seems to have accumulated as much as ~130 mm on the 1st July.

Response: We are very sorry of our mistakes in expression which let you misunderstand and we have revised the sentence in line 135-136.

13. [Line 138-139] How come the deformation increased to 11.3 mm/h? The unit!

Response: We are very sorry of our mistakes in expression and we have revised the sentence in line 137.

14. [Line 146-147 and Fig. 6] Why was the confined water not formed at the monitoring point 2 in the grave layer?

Response: The monitoring point 2 also forms confined water, but its water head is not as high as that of hole 1 because the buried depth of monitoring hole 2 is shallow.

15. [Line 147] a maximum water pressure of 26.3 m ?? Unit!

Response: We are very sorry of our mistakes in expression and we have revised the sentence in line 146-147 and Fig. 6.

16. [Line 150-151] Please theoretically justify landslide occurred bedding on the interface between the surface layer and the middle layer. If it was because of the increase in the pore-water pressure caused by confined water, isn't is more reasonable to see the sliding surface on the interface between the middle layer and the lowest layer?

Response: In line 150-159, We have explained that : on the one hand, due to the pressure of the confined aquifer, at the roof of the temporary confined water (the bottom of the silty clay layer), the "uplift pressure" perpendicular to the contact surface is generated, the force between the upper and the middle soil layers is weakened, the sliding power is increased, and the antisliding power is reduced so that the stability of the slope is reduced; on the other hand, the silty clay with gravel layer is enriched when the confined water is collected, the weight of the slope increases greatly, and the sliding power generated in the lower part of the slope also increases greatly. At the same time, under the combined action of the "uplift pressure" of the confined water and the seepage force of the middle layer, the shear failure to the initial crack point of the landslide will occur at the slope toe. This accelerates the sliding of the slope, and the uplift pressure of the temporary confined water on the soil layer interface weakens the interaction between the soil layers, thus making the slope slide along the layer.

17. [Line 155-156] "the silty clay ~ the weight of the slope increases greatly".........Again, if the authors want to insist that these unique phenomena happened, clear evidences or reasonable theoretical deduction must be suggested in advance.
Response: In line 155-156, this is a theoretical interpretation of it landslide occurred bedding on the interface between the surface layer and the middle layer.

18. [Line 157] the lower part of the slope......confusing and unclear expression. Where does it specifically indicate?
Response: As we mentioned above in line 59-61, mountains are divided into upper and lower parts.

19. [Line 168-171] Where is the exposed part of the middle layer in Fig. 4 and Fig. 7? Why no indications in those figures? Moreover, rainfall infiltration boundary is supposed to be depicted as a series of arrows. Why no such indications in those figures? Such unkind descriptions exist a lot in this manuscript.
Response: Sorry to let you misunderstand. We have revised the Fig. 1, and we showed that the upper part of the slope is usually rock with weathered fissures, which is convenient for rainfall infiltration. In addition, the exposed part of the middle layer in Fig. 4 and Fig. 7 refers to the leftmost part of the middle layer where is the inlet of rainwater infiltration.

20. [Section 3.1] There is no explanation on how the initial condition (moisture or pore-water pressure state profiles) of the slope was set using what kind of analysis, what reference field data, or what kind of different boundary conditions....
Response: The water level is shown in Fig. 7, and we added the table 2 to let you know better.

21. [Line 179 and Fig. 8] 1) "the bottom of the confining bed".......Please clarify where this expression indicate by marking the nodes in Fig. 7 where the pore-water pressure data was picked up to plot Fig. 8. 2) Why do the plots in Fig. 8 exhibit sharp conversions of the gradient from positive to negative at some distances around 30 m?
Response: We are very sorry of our mistakes in expression and we have revised the sentence in line 182.

22. [Line 186-187] "......can be divided into unconfined areas and confined areas." ........Where do the authors mean by unconfined part and unconfined part? Another unkind description....

Response: As shown in line 170-171, when the pressure head is greater than 0 m, confined water is formed. And we can see from the Fig. 8 and Fig. 9, you can distinguish unconfined areas and confined areas with 0 as the boundary.

23. [Line 190-191] What is the reason of 150 mm/d reaching the highest water level instead of 175 mm/d? Furthermore, are these results of numerical study meaningful from the perspective of the applications in practice? In what aspect? Please persuade it.

Response: The reason of 150 mm/d reaching the highest water level instead of 175 mm/d is when the rainfall reaches a certain amount, it is difficult for the rainwater to infiltrate because it's saturated.

24. [Line 198-210] It is very difficult to completely understand and match the descriptions in this paragraph with Fig. 9.

Response: We are very sorry to puzzled you and we have revised the sentence in line 200-212.

25. [Line 209] What does "lifting speed" mean?

Response: We are very sorry to puzzled you and we have revised the sentence in line 200-212.

26. [Line 218-209] the most optimal permeability 0.0001-0.0003................It is only a result deduced from a single specific event condition. What aspect does this study have valuable meaning for a general application?

Response: As described above, we use the Seep/W modular of Geo-studio software to verify the analysis that temporary confined water is the main factor triggering the landslide.

27. [Line 224] What kind of limit equilibrium theory and what equation? At which depth was the sliding surface set in the stability analysis? How did you apply the pore-water pressure term that changes over time into the limit equilibrium equation? Detailed descriptions on your assumptions and settings in the stability analysis is entirely omitted.....

Response: We are very sorry to puzzled you and we have revised the sentence in line 225-226.

28. [Line 227] Normally, we use "factor of safety" rather than "stability coefficient"....In addition, the stability coefficient is positively(?) related to the water pressure level? Not negatively?

Response: We are very sorry of our mistakes in expression and we have revised the sentence in line 229 and Fig. 11.

We tried our best to improve the manuscript and made some changes in the manuscript. These changes will not influence the content and framework of the paper. And here we did not list the changes but marked in red in revised paper.

Moreover, after we made the above modifications according to the comments of the reviewers, we found that the title of the paper can modified to " Temporary confined water responsible for triggering the landslide of a piedmont gentle slope in Zhenjing city, Jiangsu Province, China: a case study" to reflect directly the main purpose and highlight the key points of the paper.

We appreciate for Editors/Reviewers' warm work earnestly, and hope that the correction will meet with approval.

Once again, thank you very much for your comments and suggestions.

Shulan Guo
Nanjing University
Guoshulan1221@163.com

---

## Referee Comment (RC2) · Anonymous Referee #2 · 23 Mar 2020

General comment

The manuscript deals with apotentially interesting cas study of a landslide driven by the temporary build up of pressure in a gravel layer itnterbedded between the bedrock and an overlying clay layer with much smaller hydraulic conductivity. The manucript is correctly structured, the English langauge is poor, and should be revised with the help of a native speaker, but ithe manuscirpt is anyway understandable. In spite of the interesting case study, which could be of interest for part of the readership of NHESS, th manuscript suffers from many issues, related to: the description of the study case; the mathematical modeling; the interpretation of the results. All these issues make the drawn conclusions too weak, and so I regret to recommend rejection of the manuscript in its present form.

Detailed comments (more can be found in the attached annotated manuscript)

The description of the scientifiic context which this study belongs to is incomplete. The literature review is poor, and some of the cited papers seems not relevant to the studied type of landslide.

The description of the case study lacks much information: the slope is poorly described; the installed monitoring sensors are not described at all; the description of the supposed failure mechanism is confuse and unclear.

The interpretation of the monitoring results is triviial: aruing that an increaseof pore water pressure under the destabilized layer strongly contributes to the failure is obvious. What is less obvious, and should be the focus of he study, is why water easily infiltrates into the coarse layer, but it is not drained out wih the same easiness. The reason for this cannot be searched in the characteristics of the rianfall (intensity and duration), nor in the hydraulic conductivity of the gravel (in fact, it controls both infilration and drainage), but rather on some feature of the hydraulic boundary condition at the toe of the slope, which can be related to the geometry and/or to some factor external to the gravel layer. Here the authors completely miss the chance of making this study really novel and interesting, aas they completely neglect the drainag mechanism, both in the intepretation of the esults and in the (following) mathematical modelling exercis.

The mathematical model is not clearly described: the hydraulic boundary conditions are are only partially specified, and in the end it is not clear how water gets in and out of the modelling domain. Although somewhere in the paper it seems that the authors would link the release of the pressure (also 9 to the deformations of the soil, the model seems to be developed in the hypothesis of rigid soil (although also this information is not explicitly given).

The results of the model simulations are quite far from the observed evolution of the landslide: the maximum predicted pressure is half the observed, although the simulations have been carried out with rainall events much more intense than the observed, and with a conductivity of the gravel much smaller than what is declared. Given the lacking information about the model, it is difficult to understand the reason for these-large discrepancies. I guess that something about the boundary conditions, controlling the rate of infiltration and drainage from the system is completely missing in the model.

Finally, the drawn conclusions are quite obvious, and indeed it could not be different, given the neglection fo the most interesting features of the studied phenomenon. The authors come ou with some ideas for stabilization measures, also quite bvious, as they suggest to avoid too much water entering the gravel layer, and to extract water to facilitate the drainage of water already entered.

Please also note the supplement to this comment:
https://www.nat-hazards-earth-syst-sci-discuss.net/nhess-2019-404/nhess-2019-404-RC2-supplement.pdf
* * *
[Figure]

**Supplement:**

[revised manuscript text omitted]

---

## Author Comment (AC2) · 1 Apr 2020

Dear Editors and Reviewers:

Thank you for your affirmation and comments concerning our manuscript entitled "Temporary confined water responsible for triggering the landslide of a piedmont gentle slope in Ningzhen Area, China". This manuscript is very interesting, but maybe I didn't write it well. I will try my best to modify it according to your requirements and opinions. We hope the new manuscript will meet your standard. Moreover, we found that the title of the paper can modified to " Temporary confined water responsible for triggering the landslide of a piedmont gentle slope in Zhenjing city, Jiangsu Province, China: a case

study" to reflect directly the main purpose and highlight the key points of the paper.

We appreciate for Editors/Reviewers' warm work earnestly, and hope that the correction will meet with approval.

Once again, thank you very much for your comments and suggestions.

Shulan Guo Nanjing University Guoshulan1221@163.com